# Parasitism by the Tachinid Parasitoid *Exorista japonica* Leads to Suppression of Basal Metabolism and Activation of Immune Response in the Host *Bombyx mori*

**DOI:** 10.3390/insects13090792

**Published:** 2022-08-31

**Authors:** Minli Dai, Jin Yang, Xinyi Liu, Haoyi Gu, Fanchi Li, Bing Li, Jing Wei

**Affiliations:** 1School of Basic Medicine and Biological Sciences, Soochow University, Suzhou 215123, China; 2Sericulture Institute, Soochow University, Suzhou 215123, China; 3College of Plant Protection, Fujian Agriculture and Forestry University, Fuzhou 350002, China

**Keywords:** *Bombyx mori*, tachinid parasitoid, transcriptome, basal metabolism, immune response

## Abstract

**Simple Summary:**

The dipteran parasitoid Tachinidae are important biocontrol agents and some of them are pests in sericulture. We previously have demonstrated that tachinid parasitoid *Exorista japonica* parasitism causes pupation defects in *Bombyx mori*. However, the underlying mechanism is not fully understood. In this study, we performed transcriptome analysis of the fat body of *B. mori* parasitized by *E. japonica*. We found that the host basal metabolism was inhibited whereas the immune response was activated. These results indicate that the tachinid parasitoid perturbs the basal metabolism and activates the energetically costly immunity of the host, leading to the development arrest of the host. This study provides insights into how tachinid parasitoids modify host basal metabolism and immune response for the benefit of developing parasitoid larvae.

**Abstract:**

The dipteran tachinid parasitoids are important biocontrol agents, and they must survive the harsh environment and rely on the resources of the host insect to complete their larval stage. We have previously demonstrated that the parasitism by the tachinid parasitoid *Exorista*
*japonica*, a pest of the silkworm, causes pupation defects in *Bombyx mori*. However, the underlying mechanism is not fully understood. Here, we performed transcriptome analysis of the fat body of *B. mori* parasitized by *E. japonica*. We identified 1361 differentially expressed genes, with 394 genes up-regulated and 967 genes down-regulated. The up-regulated genes were mainly associated with immune response, endocrine system and signal transduction, whereas the genes related to basal metabolism, including energy metabolism, transport and catabolism, lipid metabolism, amino acid metabolism and carbohydrate metabolism were down-regulated, indicating that the host appeared to be in poor nutritional status but active in immune response. Moreover, by time-course gene expression analysis we found that genes related to amino acid synthesis, protein degradation and lipid metabolism in *B. mori* at later parasitization stages were inhibited. Antimicrobial peptides including Cecropin A, Gloverin and Moricin, and an immulectin, CTL11, were induced. These results indicate that the tachinid parasitoid perturbs the basal metabolism and induces the energetically costly immunity of the host, and thus leading to incomplete larval–pupal ecdysis of the host. This study provided insights into how tachinid parasitoids modify host basal metabolism and immune response for the benefit of developing parasitoid larvae.

## 1. Introduction

The dipteran parasitoids are important components of ecological food webs and they can be used as biological control agents against a number of crops and forest pests of economic significance [1,2]. At least 21 families of Diptera contain species with parasitoid lifestyles, and among them, the family Tachinidae are the largest, with more than 8500 described species, and have the greatest diversity [2]. Tachinids dominate the parasitoid assemblages of externally feeding larval Lepidoptera, and in some regions achieve twice the rate of parasitism of all hymenopteran parasitoids combined [3]. They have been involved in many operations of biological control against various insect pests, including *Mythimna separate*, *Helicoverpa armigera* and *Operophtera brumata*, and thus have great potential as biocontrol agents [4,5,6].

Tachinids are all endoparasitoids, mostly of larval Lepidoptera. They also attack a range of other insects, such as Coleoptera (larvae and adults), Heteroptera (nymphs and adults) and Hymenoptera Symphyta (larvae) [7]. They all have three larval instars, and can be either solitary or gregarious depending on the species [7]. The Tachinidae draw their energy and nutrient needs from hosts and are thus metabolically dependent on them. Most of the energy available in the host is used for basal metabolism (maintenance), growth and reproduction, but this process may be affected by tachinid parasitoids that increase the energy demand in order to face the stress induced. Thus, the energy stored such as protein, carbohydrates and lipids may be modified in terms of both quantity and proportions. Tachinids may indirectly increase energy expenditure of their hosts by increasing behavioral activity or activating the immune system of the host. Given the ubiquity and biological importance of tachinids, it is surprising that their impacts on hosts’ energy metabolism are still poorly understood. The tachinid parasitoid *Blepharipa sericariae* can secrete a small peptide to retard the transport of diacylglycerol from the host’s fat body [8]. We have recently demonstrated that the tachinid *E. japonica* inhibited primary sugar trehalose synthesis in the host *B. mori* [9], indicating a possible regulation of host energy and nutrition metabolism by tachinid parasitoids.

Insect defense comprises humoral and cell-mediated immunity that recognizes and kills the invading tachinid parasitoid [10]. The success of parasitization depends mainly on the ability of the host to mount an effective immune response against the parasitoid and the ability of the parasitoid to avoid or counteract this response. Unlike hymenopteran parasitoids defeating the host immune defense, tachinids can escape the host immune response by migrating out of the hemocoel or building a respiratory funnel [11,12]. In silkworms parasitized by *Exorista bombycis*, the expression of immune proteins and detoxification enzymes in hemocytes at an early infection stage was inhibited, suggesting an active suppression of hemocyte-mediated host defense [13]. Immune gene expression, melanization and apoptosis were activated in silkworm integumental epithelium post *E. bombycis* parasitization [14]. Proteomic analysis revealed that the levels of innate immune proteins and apoptosis-related proteins were induced in the hemolymph of *E. sorbillans*-parasitized silkworms at a late infection stage [15]. These findings demonstrated that different host tissues exhibit diverse responses to tachinid parasitoids. Generally, the tachinid fly lays eggs on the host integument or food plant, then the hatched larvae invade from host integument or intestine into the hemocoel and build respiration funnels, the developing tachinid larvae are directly immersed in the hemolymph and contact with host hemocytes and fat body [16]. Therefore, tachinid parasitization should have profound effects on the physiology of host integument, hemocytes and fat body. The insect fat body is not only a metabolic organ used for storing energy and providing energy and nutrients needed for growth and development, but also an organ important for innate immune responses [17,18]. However, the response mechanism of the insect fat body to tachinid parasitoid remains unclear.

The tachinid, *E. japonica*, a larval parasitoid of Lepidoptera, can attack larvae of around 18 lepidopteran families, mainly Lymantriidae, Lasiocampidae, Noctuidae and Arctiidae, and thus can be exploited as regulators of target insect pests [19]. *E. japonica* oviposits macrotype eggs on the host integument. The newly hatched larvae penetrate into the host body and continuously develop until pupation, which occurs outside the host’s larval remains [20]. We previously demonstrated that *E. japonica* dysregulated the biosynthesis and signaling of 20-hydroxyecdysone (20E) and trehalose synthesis in the host *B. mori*, which disturbed the process of host larval–pupal transition [9]. The lipid, cholesterol, is the biosynthetic precursor to 20E, the fact that *E. japonica* parasitization induced inhibition of 20E and trehalose synthesis reveals regulation of host energy and nutrient metabolism. At present, it is unknown how tachinid parasitoid manipulates those host metabolic processes. In this study, the fat body of *B. mori* at 4 days after parasitization (*B. mori* entered the wandering stage) was used for transcriptome analysis to further explore the mechanisms in tachinid–host interaction. A number of genes involved in primary metabolism, development and defense responded to tachinid parasitization. The results provide a comprehensive view of the molecular response to tachinid parasitoid parasitization in its lepidopteran host and contribute to a better understanding of host–dipteran parasitoid interactions.

## 2. Materials and Methods

### 2.1. Insect Rearing

The silkworm (Jing song) larvae were reared with fresh mulberry leaves at 26 ± 2 °C, 60–80% relative humidity and a photoperiod of 14:10 h (L:D). The laboratory colonies of adult *E. japonica* were maintained with 20% honey solution at the same environmental conditions as silkworm larvae. Fifth instar silkworms were provided as the egg-laying host for the mated female tachinids. 

### 2.2. Parasitization of B. mori by E. japonica

For the parasitization treatment, ten 5th instar larvae in a transparent plastic box (20 cm × 10 cm × 30 cm) reared with sliced mulberry leaves were exposed to 3–5-day-old mated female *E. japonica* adults. The female *E. japonica* exhibited a featured parasitization behavior in which it stung the ovipositor on the host integument for several seconds, and once the parasitization behavior was observed we collected the silkworm larva and reared it with mulberry leaves. At 3 days after egg-laying, a visible, black-marked respiratory funnel on the silkworm integument appeared that indicated invasion of the tachinid larva. For gene expression analysis, the fat body of parasitized hosts was dissected and collected at 3, 4, 5, 7 and 8 days after *E. japonica* larva invasion into the host hemocoel. Fat bodies collected from nonparasitized host larvae at the same development period were taken as controls. Ten fat body samples (sex ratio F/M = 1:1) from parasitized or nonparasitized host larvae were pooled as one biological sample, and each treatment had three replicates. The fat body samples were then stored at −80 °C for further transcriptome and gene expression analysis.

### 2.3. Transcriptome Analysis

Total RNA of fat body samples from silkworms at 4 days after parasitization or nonparasitized host larvae was extracted with TRIzol (Life Technologies, Carlsbad, CA, USA) and purified with the RNeasy Mini Kit (QIAGEN, Hilden, Germany). RNA quality was assessed with a Bioanalyzer 2100 (Agilent Technology, Santa Clara, CA, USA). The libraries were constructed using a TruSeqTM RNA Sample Prep Kit (Illumina, San Diego, CA, USA) following the manufacturer’s instructions and sequenced on an Illumina NovaSeq 6000 platform, and 150 bp paired-end reads were generated. Raw reads were processed using Trimmomatic (version: 0.36, Leading: 3, Trailing: 3, Sliding window: 4:15, Minlen: 75). Adaptors and low-quality reads were removed to obtain clean reads. The clean reads were mapped to the reference *B. mori* genome using Hisat2 tools with default settings. A matrix of raw counts per gene was generated using “featureCounts” from the Rsubread package (version 1.30.5, http://www.bioconductor.org/packages/release/bioc/html/edgeR.html/, Liao Yang, Australia). Differentially expressed genes were identified using edgeR. The gene expression levels were measured and normalized as FPKM (fragments per kilobase of transcript, per million fragments sequenced). The transcripts with fold changes >1 and false discovery rate (FDR) adjusted *p* value < 0.05 were considered as significantly differentially expressed.

### 2.4. Gene Ontology (GO) and Kyoto Encyclopedia of Genes and Genomes (KEGG) Enrichment Analysis

The GO enrichment analysis method was conducted using GOseq R package, which is based on Wallenius non-central hypergeometric distribution, and can adjust for gene length bias in DEGs. KEGG is a bioinformatics database for the systematic analysis of gene function (http://www.genome.jp/kegg/, accessed on 15 August 2021) [21]. Significant pathway enrichment analysis was conducted with KEGG pathways as units, and hypergeometric tests were used to identify significant enrichment pathways by using KOBAS 2.0 software (http://kobas.cbi.pku.edu.cn/kobas3, accessed on 15 August 2021). The *p* value was calculated by Bonferroni correction, and a *p* value < 0.05 was considered as significant enrichment.

### 2.5. Reverse Transcription-Quantitative PCR (RT-qPCR) Analyses

Candidate transcriptomic genes were validated by RT-qPCR analysis using three biological replicates. Gene-specific primers (Appendix A) were designed using NCBI Primer-blast and synthesized commercially (Sangon Biotech, Shanghai, China). First-strand complementary DNA was synthesized using SuperScript III Reverse Transcriptase (Invitrogen, Carlsbad, CA, USA) in accordance with the manufacturer’s instructions and reverse transcribed into cDNA. RT-qPCR was performed on a Viia 7 Real-time PCR System (Applied Biosystems, Foster City, CA, USA) with SYBR Premix Ex TaqTM II (TaKaRa) for 2 min at 95 °C, followed by 40 cycles for 5 s at 95 °C and 35 s at 60 °C. Relative quantification of gene expression was calculated and normalized using ribosomal protein 49 (*Rp49*) as an internal standard. The dissociation curve was used to confirm the specificity of the primers. The 2^−ΔCt^ method was used to calculate the fold changes in gene expression level.

### 2.6. Statistical Analysis

The relative expressions of basal-metabolism- and immune-response-related genes at different time points within each group were analyzed using one-way ANOVA followed by Tukey’s test. A *p* value of < 0.05 *, 0.01 ** or 0.001 *** was considered significant, highly or the most highly significant, respectively. All analyses were conducted in using SPSS 19.0 (SPSS, Chicago, IL, USA). Figures were drawn using GraphPad Prism 7 and assembled in Adobe Illustrator CS6.

## 3. Results

### 3.1. Global Transcriptomic Changes in the Fat Body of B. mori after E. japonica Parasitization

To comprehensively understand the response of *B. mori* to *E. japonica* parasitization, we performed RNA-seq analysis of the fat body from parasitized silkworm larvae at 4 days after parasitization (DAP) when the host had dramatic changes in physiology at the wandering stage, and the fat body from nonparasitized *B. mori* larvae at the same age served as the control (Figure 1A). Three independent biological replicates of each condition were sequenced. By using Pearson’s correlation coefficient and hierarchical cluster analysis, high reproducibility was found in the samples of each condition (Figure 1B). The total number of raw reads obtained from the six samples ranged from 45.63 to 59.41 million (Appendix A). In total, 41.22–48.71 million clean reads were obtained after removal of the low-quality nucleotides (Appendix A). All the quality Q30 values (sequencing error rate < 0.1%) of the six samples were larger than 94.24%, indicating the high quality of the RNA-Seq data. Meanwhile, 90.03–91.57% clean reads could be mapped to the *B*. *mori* reference genome (SilkDB 3.0, https://silkdb.bioinfotoolkits.net, accessed on 12 August 2021). Differential gene expression analysis identified altered gene expression in 1361 genes (fold change ≥ 1, *p* < 0.05) between nonparasitized controls (NP) and *E. japonica* parasitized hosts (P), with 394 genes up-regulated and 967 genes down-regulated (Figure 1C). To validate the RNA sequencing data, we technically confirmed the expression pattern of a subset of genes involving in nutrient metabolism and immunity by RT-qPCR measurements. All of the ten tested genes exhibited similar transcription patterns when analyzed by RT-qPCR and RNA sequencing, indicating that alterations in gene expression are valid and not biased by the experimental approach (Figure 1D).

### 3.2. Functional Annotation of DEGs and Pathway Enrichment Analysis

To uncover the functions of DEGs induced by *E. japonica* parasitization, we identified functional categories of DEGs using GO analysis, and enrichment analysis was used for KEGG pathways. Among the 1361 DEGs, 624 DEGs were assigned GO terms from the three main categories: biological process (420 DEGs), cellular component (554 DEGs) and molecular function (349 DEGs) (Figure 2). The remaining DEGs failed to obtain a GO term largely due to their uninformative (e.g., ‘unknown’ or ‘uncharacterized’ protein) descriptions. In the ‘biological process’ category, DEGs were mainly located in ‘growth’, ‘immune system process’, ‘interspecies interaction between organisms’, ‘response to stimulus’, ‘developmental process’, ‘biological regulation’, ‘metabolic process’, ‘cellular process’, etc. Within the ‘cellular component’ category, ‘protein-containing complex’ and ‘cellular anatomical entity’ were predominantly represented. In the ‘molecular function’ category, the two most abundantly represented terms were ‘binding’ and ‘catalytic activity’. 

To further characterize the expression changes in parasitized hosts, we performed KEGG pathway enrichment analysis to assign biological pathways to the DEGs. Using the criteria of *p* value < 0.05, the up-regulated DEGs were significantly enriched in 7 pathways related to the immune system, endocrine system, signal transduction and amino acid metabolism, including ‘antigen processing and presentation’, ‘estrogen signaling pathway’, ‘isoquinoline alkaloid biosynthesis’, ‘MAPK signaling pathway’, ‘protein processing in endoplasmic reticulum’, ‘indole alkaloid biosynthesis’, and ‘arginine biosynthesis’. The down-regulated DEGs were significantly enriched in 14 pathways related to the metabolism of energy, lipid and carbohydrate, including ‘oxidative phosphorylation’, ‘thermogenesis’, ‘fatty acid biosynthesis’, ‘biosynthesis of amino acids’, ‘carbon metabolism’, ‘pyruvate metabolism’, etc., (Table 1). This finding demonstrated that the homeostasis of host energy and nutrition, and immunity were modulated, presumably in correspondence with the most demanding part of *E. japonica* larval growth. 

### 3.3. Down-Regulation of Genes Involved in Host Energy and Nutrient Metabolism

Either the host or the tachinid parasitoid utilizes carbohydrates, lipids and proteins or amino acids as energy resources, which raises the possibility that *E. japonica* larva might compete for these nutrients with the host. Most of the DEGs in the host fat body were enriched in these metabolic pathways, indicating that the tachinid parasitoid affected the host’s energy and lipid metabolism. In the KEGG category, ‘energy metabolism’, the transcription of 16 genes that were mainly involved in oxidative phosphorylation and nitrogen metabolism (2 genes) was decreased, whereas only one gene showed up-regulated expression (Table 2), suggesting that tachinid parasitization disrupted energy homeostasis in the host. 

Regarding amino acid metabolism, five DEGs, including two up-regulated and three down-regulated genes were significantly enriched in arginine biosynthesis. Intriguingly, the expression of nitric oxide synthase was increased by 4.8-fold, indicating the involvement of nitric oxide signaling in host responses to parasitization. The expression of the key enzyme in cysteine and methionine metabolism, malate dehydrogenase 1, was down-regulated 7.5-fold (Table 2). Meanwhile, we explored the expression patterns of key genes in amino acid synthesis and protein degradation in *B. mori* following *E. japonica* parasitization by RT-qPCR. The amino-acid- and protein-synthesis-related genes, 40S ribosomal protein SA (*RpSA*), showed decreased transcription at 7-DAP, whereas ribosomal protein P0 (*RpP0*) showed up-regulation at 5-DAP but decreased at 3- and 7-DAP (Figure 3A). The protein-degradation-related genes, *B. mori* cysteine proteinase (*Bcp*), showed an up-regulated expression at 3- and 5-DAP but a down-regulation at 7- and 8-DAP, whereas serine carboxypeptidase (*Scp*) was significantly up-regulated at 8-DAP (Figure 3A). It is likely that tachinids can modify the amino acid and protein balance of their hosts to meet their developmental requirements.

Carbohydrates that can be de novo synthesized via gluconeogenesis from amino acids are the primary resource of sugar for trehalose synthesis in insects [22]. In parasitized hosts, we identified 19 DEGs involved in carbohydrate metabolism in *B. mori*, with 6 genes up-regulated and 13 genes down-regulated (Table 2). Particularly, all down-regulated genes were associated with glycolysis/gluconeogenesis, pyruvate metabolism and glyoxylate and dicarboxylate metabolism. Furthermore, for the glycolysis pathway, seven genes including glucose-6-phosphate isomerase (*Pgi*), phosphofructokinase (*Pfk*), triose phosphate isomerase (*Tpi*), glyceraldehyde 3-phosphate dehydrogenase (*Gadph*), phosphoglyceromutase (*Pglym*), enolase (*Eno*) and lactate dehydrogenase (*Ldh*) showed significantly decreased transcription at 3- and 5-DAP, and most of them were still down-regulated at 7- and 8- DAP (Figure 3B). These data suggest an inhibitory effect of parasitism on the carbohydrate metabolism of the host, thus leading to reduced trehalose synthesis as we demonstrated previously [9].

In addition, 10 DEGs were involved in lipid metabolism, with 7 genes down-regulated and 3 genes up-regulated (Table 2). Specifically, the transcription of fatty acid synthase (FAS) and fatty acyl-CoA reductase-like genes was decreased by 5-fold and 10-fold, respectively, suggesting inhibition of lipogenesis in the host. The transcription of phospholipase A2, which is thought to be the first step in eicosanoid biosynthesis, was up-regulated 6.14-fold. Two uridine diphosphate glucosyltransferase (UGT) genes, UGT48C1 and UGT2, which are detoxification enzymes involved in the biotransformation of various lipophilic endogenous compounds and xenobiotics, showed 5- and 112-fold up-regulation, respectively. We further determined that the expression levels of key enzyme genes involved in lipid biosynthesis, including phospholipase C β1 (*PLC-β1*), diacylglycerol O-acyltransferase 1 (*DGAT1*), cardiolipin synthase (*CRLS1*), alkaline ceramidase 3 (*ACER3*), sphingomyelin synthase 1 (*SGMS1*) and ceramide synthase (*CERS3*), were all significantly down-regulated in *B. mori* at 3- and 7-DAP. At 5-DAP, the transcriptional level of *ACER3* was up-regulated whereas other genes did not respond. The transcriptional levels of *PLC-β1*, *DGAT1*, *ACER3*, *SGMS1*, and *CERS3* were persistently reduced at 8-DAP (Figure 3C).

### 3.4. Regulation of Host Development-Related Genes

We have previously demonstrated that *E. japonica* can regulate silkworm development through targeting JH and 20E activities. Specifically, 20E titer in hemolymph was increased, whereas JH titer was decreased in parasitized silkworms [9]. In this study, when comparing the nonparasitized and parasitized silkworms, two juvenile hormone binding protein (JHBP) genes, which are important for the transportation of JH to its target tissue and to prevent JH degradation by generalist esterases, showed down-regulation in the fat body of parasitized silkworms (Table 3). Juvenile hormone esterase (JHE) was expressed at a much lower level in the parasitized insects, consistent with our previous results [9]. Meanwhile, in the parasitized groups, ecdysteroid-phosphate phosphatase (EPPase), a key enzyme for the conversion of inactive ecdysteroid-phosphates to active ecdysteroids, showed a 3.9-fold up-regulation, which logically led to an increase of 20E in hemolymph as we demonstrated previously [9]. Additionally, the expression of ecdysteroid-regulated 16 kDa protein was significantly down-regulated.

### 3.5. Manipulation of Host Cellular Immune Responses by E. japonica Parasitization

The insect hosts commonly protect themselves from parasitoid invasion by cellular defenses including phagocytosis, encapsulation and nodulation [10]. These processes involve pattern recognition receptors and immune effectors [23]. In the fat body of parasitized *B. mori*, hemolin, which mediates phagocytosis and nodulation in insects challenged by bacteria or viruses, was significantly up-regulated (Table 4). However, *hemolin* was down-regulated at 3-, 7- and 8-DAP (Figure 4A). *Hemocytin*, a major mediator of nodule formation, had down-regulation at 3-, 4- and 5-DAP, whereas it was up-regulated at 7- and 8-DAP (Table 4, Figure 4A). C-type lectin (*CTL11*), which can bind to larval hemocytes and various pathogen-associated molecular patterns to enhance the hemocyte-mediated immune response, was highly induced in the silkworm at 3- and 5-DAP (Figure 4A). Moreover, at 4-DAP, two scavenger receptors, type B and C precursor, involving in phagocytosis of exogenous materials, were down-regulated by about 3-fold (Table 4). Integrin β3 and β4, which are important for phagocytosis of microorganisms by insect hemocytes, were differentially regulated (Table 4). In addition, the process of host phagocytosis involves plasma membrane repair and cytoskeleton rearrangement to eliminate foreign organisms [24]. Intriguingly, among the cytoskeletal regulatory DEGs in the fat bodies of parasitized *B. mori*, except the up-regulation of heat shock protein 90, actin3 and stathmin, most notable was a down-regulation in the host of >40 cytoskeletal regulatory genes (Table 4). These results indicate that the cellular response in the host was modulated by *E. japonica*.

### 3.6. Induction of Humoral-Immune-Response-Related Genes

#### 3.6.1. Melanization

Melanization is initiated in insects to kill and eliminate invasive pathogens or parasites followed by recognition of microbial elicitors by pattern recognition receptors such as β-1,3-glucan (βGRP), lipopolysaccharide and peptidoglycan [25]. In the parasitized host fat body, we observed increased expression of βGRP3. Additionally, the expression of three peptidoglycan recognition protein genes (PGRPs), including PGRP-LB-like, PGRP-S2 and PGRP-S6 was strikingly increased by 76- to 2949-fold, respectively, indicating recognition of parasitoid by pattern recognition receptors in the host (Table 5). After recognition, the transcription levels of two serine protease genes were up-regulated 42- and 175-fold, suggesting a super activation of the serine protease cascade melanization pathway in the host. Meanwhile, two negative regulators of melanization, serine protease inhibitors (serpins) and angiotensin converting enzymes (ACEs) showed distinct expression patterns in parasitized hosts. Two ACE genes showed a 4- and 16-fold decrease, whereas three serpin genes were expressed at 2.5-, 3- and 43-fold higher levels in parasitized hosts, implying that the host probably modulated ACEs and serpins to prevent itself from excessive melanization, or the parasitoid might produce factors to manipulate their expression, inhibiting the host melanization pathway.

#### 3.6.2. Antimicrobial Peptides

The production of antimicrobial peptides (AMPs) is the major feature of the humoral immune response in insects [26]. In this study, the expressions of 13 antibacterial peptide genes including the lysozyme, *gloverin*, *cecropin*, *enbocin*, *attacin* and *moricin* families were all up-regulated in response to tachinid parasitoid attack (Table 5). Specifically, the transcriptional level of *moricin* was dramatically increased by 72-fold, and other AMP genes showed a 3- to 26-fold increase in expression. The time-course gene expression analysis showed that, the transcriptional levels of *CecA*, *Gloverin1*, *Gloverin2* and *MoricinB3* were significantly up-regulated at 3-DAP. In addition, the expression levels of *CecA*, *Gloverin1*, *Gloverin2*, *Moricin* and *MoricinB3* of the host were significantly increased by 1.74- to 15.37-fold at 5-, 7- and 8-DAP (Figure 4B).

#### 3.6.3. Immune-Related Signaling Pathways

In insects, signal transduction pathways, including the Toll, immune deficiency (IMD), mitogen-activated protein kinases (MAPK), Janus kinase/signal transducers and activators of transcription (JAK/STAT) and the transforming growth factor-beta (TGF-β) pathways, coordinate to mediate immune responses [27,28]. Generally, the inducible expression of AMP genes mainly depends on stimulation of Toll and IMD signaling pathways in insects [29]. The induction of beta-1,3-glucan recognition protein3 and AMPs, and the up-regulation of a *Drosophila* pirk homolog, which has been reported as a negative regulator of IMD pathway in *Drosophila* [30], were observed, suggesting that these two pathways responded to *E. japonica* parasitism (Table 5). We further noticed that the transcriptional level of Toll signaling pathway gene, *Spatzle*, was significantly decreased at 3- and 8-DAP, whereas *Cactus*, a negative regulator of the Toll pathway, was significantly up-regulated at 7- and 8-DAP, suggesting that the Toll pathway was inhibited in parasitized *B. mori*. The IMD signaling pathway gene, *Relish*, exhibited decreased expression at 5-DAP but showed an up-regulation at 8-DAP, and *Imd* was down-regulated at 5-DAP but up-regulated at 3- and 8-DAP (Figure 4C). Moreover, in parasitized hosts, one positive acting gene in the JAK/STAT pathway, signal transducing adapter molecule 2 (STAM2), exhibited about a 2-fold increase in transcription (Table 5). Quantitative analysis showed that the JAK-STAT signaling pathway gene, *Hop*, was up-regulated at 8-DAP, and *Stat* was significantly down-regulated at 3-DAP but increased at later time points, suggesting that the JAK-STAT pathway was inhibited at an early infection stage but activated at a late infection stage. In addition, the MAPK pathway genes, stathmin, heat shock protein 68, hsp70 and growth arrest and DNA damage-inducible protein GADD45α, were up-regulated by 2- to 52-fold, whereas the epidermal growth factor receptor (EGFR), which activates MAPK pathways via binding to its ligands, showed down-regulation after parasitization (Table 5). TGF-β signals are transduced into the nucleus of the cell by the transcription factor Smad [31]. The TGF-β ligand, the bone morphogenetic protein (BMP)-type gene glass bottom boat (gbb), was significantly over-transcribed in parasitized hosts compared with the control ones. We further found that Tigf, a Smad co-repressor that negatively regulates TGF-β signaling was down-regulated [32], suggesting that the TGF-β pathway in *B. mori* was activated by parasitoid fly infection. These data indicate that the signaling pathways, Toll and IMD, JAK/STAT, MAPK and TGF-β in *B. mori* might be involved in parasitization-induced stress responses.

## 4. Discussion

Few studies have analyzed the responses of hemocytes, integument and hemolymph of lepidopteran hosts such as *B. mori*, *M. separate* and *Galleria mellonella* to tachinid parasitoids [4,9,11]. The insect fat body is the major organ where various phenomena such as storage, synthesis and degradation occur systematically and serves as the main storage organ for the nutrient reserve, such as carbohydrate, lipid and protein, and is an immune responsive tissue [18]. To reveal more details in the response of host fat body to tachinid parasitization, gene expression profiles of the fat body of *E. japonica*-parasitized *B. mori* at the wandering stage were investigated in this study. Our results indicate that the expressions of 1371 genes were regulated as a consequence of parasitization. These DEGs were predominant in energy and nutrient metabolism as well as immune response. 

The insect parasitoid larvae completely depend on host-derived energy and nutrients for growth and development [33]. During their long-term co-evolution, parasitoid larvae have evolved a variety of ways to manipulate their host’s physiology to increase nutrient availabilities and accumulate energy reserves, which requires dramatic adjustments of the metabolic strategy employed by parasitoids [34]. For example, hemolymph trehalose levels of *Heliothis virescens* and *Trichoplusia ni* following parasitization by *Microplitis croceipes* and *Hyposoter exiguae,* respectively, were elevated [35]. Parasitization by the hymenopteran, *Euplectrus separatae*, results in a release of fat particles from the host’s fat body and an increase in hemolymph free fatty acids of the host [36]. Parasitism also induces changes in the amount of amino acids, proteins, pyruvate and carbohydrates within the host in both endo- and ectoparasitoids [37,38]. In the current study, amino acid and protein synthesis was inhibited at most of the tested time points after parasitization, as revealed by the down-regulated expression of *RpSA* and *RpP0*; meanwhile, *Bcp* and *Scp*, which mediate protein degradation, were up-regulated, which was probably due to the manipulation of the host protein degradation pathway by the parasitoid larvae to ensure amino acid availability. The hymenoptera parasitoid *Cotesia vestalis* can suppress host lipogenesis and reduce the systemic lipid level of the host [39]. Similarly, we found that the expression levels of lipid-synthesis-related genes of *B*. *mori* were reduced after *E*. *japonica* parasitism, suggesting that a conserved mechanism for regulating host lipid biosynthesis occurred in both hymenopteran and dipteran parasitoids. Hymenoptera parasitoids activate the host immune response and that is accompanied by a metabolic shift that results in up-regulation of specific glycolytic enzyme genes in the fat body to produce much ATP [40]. In contrast, we show that the glycolysis in the *B. mori* fat body was inhibited after parasitization, implying that insect parasitoids with different oviposition strategies or parasitization strategies may modulate host energy metabolism in diverse ways. Indeed, direct uptake of nutrients and utilization of energy from the host tissue is highly advantageous for parasitoid larvae because they can avoid the substantial metabolic costs that are required for development. Therefore, changes in energy and nutrient metabolism in the host to meet the required energy demands of parasitoids can be expected.

Upon detection of parasitoid infection, the host immune system aims to defend against and clear the infection. Cellular immune responses mediated by hemocytes including nodulation, encapsulation, phagocytosis and humoral responses such as melanization and secretion of antimicrobial peptides are activated to precede final clearance of the invading parasitoids from the hemolymph [10]. Meanwhile, parasitoids must overcome the inhospitable environment of host hemolymph to complete their larval development. They have evolved to adapt to survival owing to the ability to avoid the host’s immune response. For example, the hymenopteran parasitoids can inject polydnavirus and venom to suppress the host’s immune response [41]. The dipteran parasitoids such as *B. sericariae*, *Zenillia libatrix* and *Drino inconspicuoides* can avoid immune responses in hemolymph through migrating into the ganglion, abdominal muscles or silk glands of the host or building a respiratory funnel from products of the host’s immune response [8,11]. For resisting Tachinidae attack, the host cellular responses, such as phagocytosis, should be completed in less than 48 h, because at that time the 1st instar parasitoid has developed to 2nd instar that likely becomes more reactive to suppress host defense. In our study, at 4-DAP, the transcription of most genes related to nodulation and phagocytosis in *B. mori* was suppressed, indicating that phagocyte deficiency, cytoskeleton degradation and adhesion disruption occurred in infected hosts. Although *hemolin* and *hemocytin* were up-regulated at 7- and 8-DAP, the host’s immune system was unable to eliminate the parasitoid. 

The main characteristics of insect humoral immunity include secretion of antimicrobial peptides, which is considered as the first line of defense against the invasion of pathogenic microorganisms [42,43]. In addition to Toll and IMD pathways, a conserved signaling pathway that is known to be associated with the production of AMPs, the insulin-like signaling (ILS) pathway, has been recently reported to be involved in the regulation of AMPs in *Drosophila melanogaster* and *B. mori* [44]. The expression of the AMP, diptericin, was induced after infection by parasitoid wasps, which the authors inferred was associated with the encapsulation of wasp eggs [45]. There is growing evidence that parasitoid infestation up-regulates the expression of several AMP genes in the host, thereby enhancing resistance to parasitization [9,46,47]. Intriguingly, our results exhibited that the expression levels of AMP genes in parasitized *B. mori* were up-regulated at most of the time points that we measured, which could be interpreted as contributing to overall activation of the humoral response. 

The Toll, IMD and JAK/STAT signaling pathways act as the core portion of silkworm humoral immunity to infection. Deletion of genes in Toll pathway leads to failed encapsulation of wasp eggs [48]. In this study, after parasitism by *E. japonica*, the transcriptional level of Toll signaling pathway gene, *Spatzle*, was down-regulated, and *Cactus*, the negative regulator of this pathway, maintained up-regulated expression at the late infection stage. These results suggested that parasitism exerted an inhibitory effect on the Toll pathway in *B. mori*, which presumably increased the risk of parasitoid infection. Gene mutation in IMD pathway weakened the resistance to pathogens in insect hosts. IMD-pathway-related immune defense genes were highly expressed in *Drosophila* when parasitized by *Asobara tabida* or *Leptopilina boulardi* [49]. Similarly, in parasitized *B. mori* the transcriptional level of *Relish* was down-regulated at 5-DAP and significantly increased at late parasitization, meanwhile, *Imd* was up-regulated at 3- and 8-DAP. It is therefore possible that the up-regulation of IMD pathway genes was responsible for increased AMP expression, contributing to resisting tachinid parasitoid attack. The parasitoid wasp infection activated JAK/STAT signaling in *Drosophila* larvae, and the interruption of JAK/STAT pathway increased the survival rate of wasps [46,50]. We showed that the transcriptional levels of *Hop* and *Stat* in parasitized silkworms were up-regulated at the late infection stage, indicating that *E. japonica* parasitism infection probably induced the JAK/STAT pathway in silkworm at a late infection stage. Indeed, immune response of host against parasitoid is a complex and time-associated process, and the information on precisely when cellular or humoral immunity was mounted remains unclarified. Thus, further studies would be required to pinpoint the timing of immunity of the different immune mechanisms that govern the outcome of tachinid parasitization.

Immune activation is energetically costly and impairs an insect’s ability to acquire the resources it needs to support basal metabolism. It has been suggested that energetic resources are indeed reallocated, perhaps from stored reserves, to support immune system activity [51]. Particularly, amino acids are necessary for the structure of immune pathway peptides and effectors such as AMPs. Dietary carbohydrates and lipids supply the energy needed for metabolic actions in both humoral and cellular immune responses. For example, *Drosophila* larval skeletal muscles can affect cellular immune response against wasp infection by controlling carbohydrate metabolism [52]. The steroid hormone 20E and the sesquiterpenoid juvenile hormone (JH) are both involved in the regulation of the inducibility of AMP genes and the IMD-dependent responses in insects [53]. In this study, the tachinid parasitoid inhibited host basal metabolism, which resulted in insufficient energy supply for host development and immunity. AMP production relies on protein resources and steroid hormone 20E synthesis depends on lipid metabolism, regarding to the limited host resources, more energy and resources might be spent to maintain the persistently high induction of AMP gene transcription and 20E synthesis in parasitized hosts, whereas other genes that affect developmental traits and immune strategies were probably unable to be activated persistently. The interactions between basal metabolism and immunity in the host attacked by tachinid parasitoids should be examined in the future work. 

## 5. Conclusions

Tachinid parasitoid *E. japonica* parasitization triggered tremendous changes in basal metabolism and immunity of the host *B. mori*. Basal metabolic pathways were mostly inhibited after the parasitization, including energy metabolism, carbohydrate metabolism, amino acid metabolism and lipid metabolism, etc. The host immune responses, including the cellular and humoral immune response were also modulated. This study extends our knowledge of the molecular interactions between dipteran parasitoids and the host. Further detailed mechanistic studies should investigate how parasitoid survival is achieved via manipulation of host basal metabolism and immunity.

## Figures and Tables

**Figure 1 insects-13-00792-f001:**
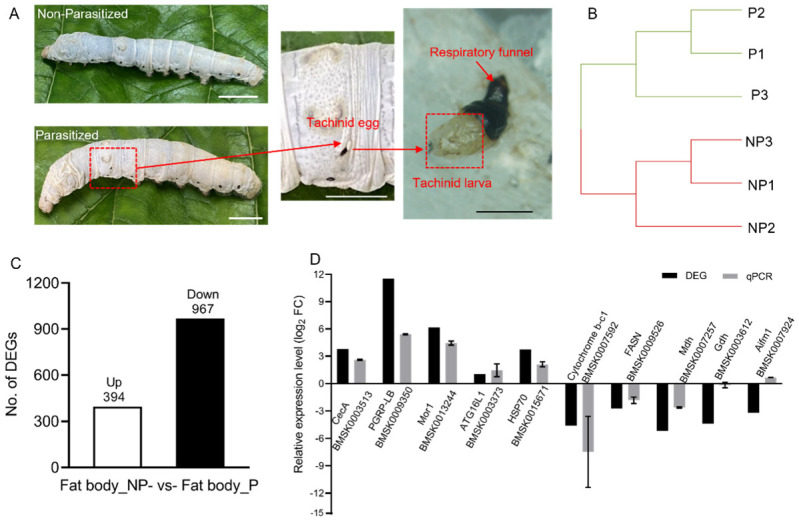
Global transcriptomic changes in the fat body of parasitized *B. mori*. (**A**) The parasitized and nonparasitized *B. mori* and an *E. japonica* larva with a respiratory funnel developed within the host. White bars, 1 cm; black bar, 100 μm. (**B**) A cluster analysis of 6 samples. P, the parasitized silkworms; NP, nonparasitized controls. (**C**) Number of DEGs after *E. japonica* parasitism. Up, up-regulated genes. Down, down-regulated genes. (**D**) Validation of ten DEGs by RT-qPCR analysis. *CecA*, cecropin A; *PGRP-LB*, peptidoglycan recognition protein-LB; *Mor1*, moricin1; *ATG16L1*, autophagy-related protein 16-like; *HSP70*, heat shock protein 70; *FASN*, fatty acid synthase; *Mdh*, malate dehydrogenase; *Gdh*, glutamate dehydrogenase; *Aifm1*, apoptosis-inducing factor isoform X1. All transcriptional data were normalized to the expression level of *Rp49*.

**Figure 2 insects-13-00792-f002:**
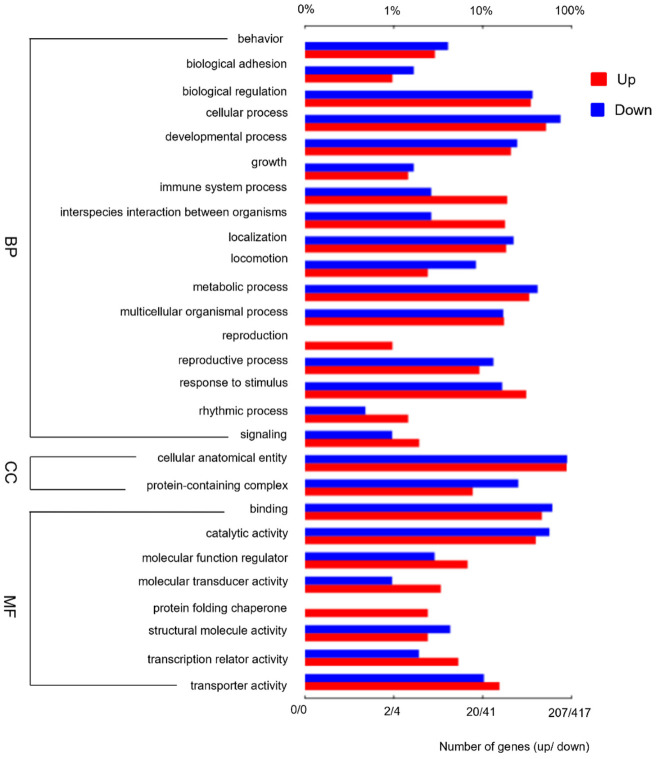
Summary of gene ontology (GO) classification of differentially expressed genes in *B. mori*. The upper top *x*-axis represents the proportion of the percentage of the number of genes corresponding to the function, and the lower *x*-axis represents the number of genes corresponding to the function. Up, up-regulated genes; Down, down-regulated genes. BP, biological process; CC, cellular component; MF, molecular function.

**Figure 3 insects-13-00792-f003:**
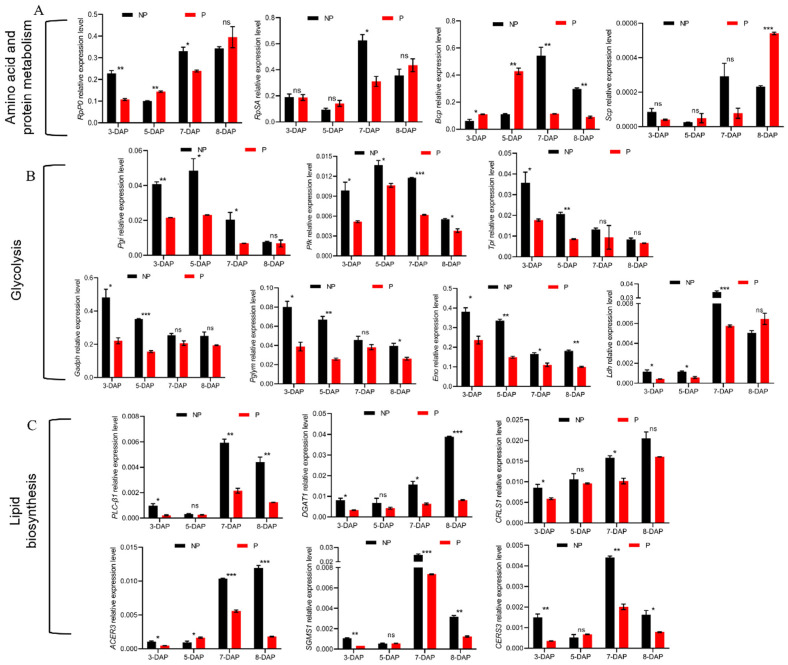
Effects of *E. japonica* parasitism on the expression levels of genes involved in amino acid and protein metabolism, carbohydrate metabolism, and lipid biosynthesis of *B. mori* at 3-, 5-, 7- and 8-DAP. (**A**) The relative expression levels of genes involved in amino acid and protein metabolism. (**B**) Expression levels of metabolic genes involved in glycolysis pathway. (**C**) RT-qPCR analysis of the expression levels of key enzymes involved in lipid biosynthesis. All transcriptional data were normalized to the expression level of *Rp49*. The statistical significance is indicated by * *p* < 0.05, ** *p* < 0.01 or *** *p* < 0.001. The results are shown as the mean ± S. D.

**Figure 4 insects-13-00792-f004:**
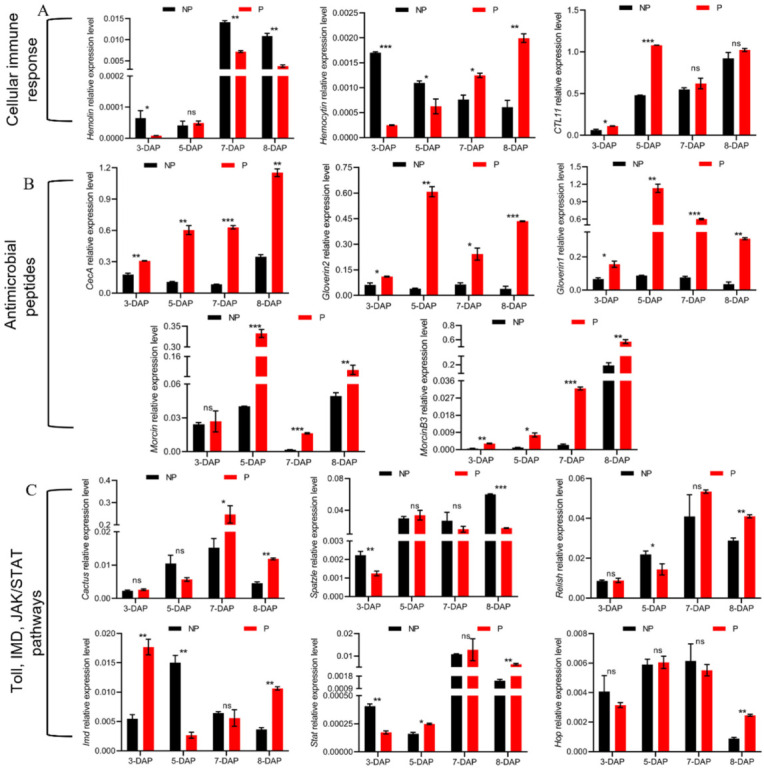
Transcriptional characteristics of immune-related genes induced by *E. japonica* parasitism. (**A**) The relative expression levels of cellular-immunity-related genes. (**B**) The relative expression levels of AMP genes. (**C**) The relative expression levels of Toll, IMD pathway genes and JAK/STAT pathway genes. All transcriptional levels were normalized to the expression level of *Rp49*. The statistical significance is indicated by * *p* < 0.05, ** *p* < 0.01 or *** *p* < 0.001. The results are shown as the mean ± S.D.

**Table 1 insects-13-00792-t001:** KEGG pathway enrichment analysis of DEGs in *B. mori* after parasitization.

Category	KEGG Term	KO ID	Input/Background Number	*p* Value
Up-regulated DEGs	Antigen processing and presentation	ko04612	7/24	1.19 × 10^−5^
	Estrogen signaling pathway	ko04915	6/38	0.001819
	Isoquinoline alkaloid biosynthesis	ko00950	2/5	0.011286
	MAPK signaling pathway	ko04010	6/60	0.017507
	Protein processing in endoplasmic reticulum	ko04141	9/122	0.026165
	Indole alkaloid biosynthesis	ko00901	1/1	0.034861
	Arginine biosynthesis	ko00220	3/23	0.044288
Down-regulated DEGs	Oxidative phosphorylation	ko00190	34/131	8.86 × 10^−11^
	Phagosome	ko04145	15/65	7.89 × 10^−5^
	Thermogenesis	ko04714	28/177	0.000146
	Cardiac muscle contraction	ko04260	11/42	0.000216
	Gap junction	ko04540	10/41	0.00077
	Valine, leucine and isoleucine biosynthesis	ko00290	4/7	0.000961
	Carbon metabolism	ko01200	16/144	0.011988
	Biosynthesis of amino acids	ko01230	11/69	0.014215
	Fatty acid biosynthesis	ko00061	5/21	0.018282
	Pyruvate metabolism	ko00620	6/32	0.031156
	Cutin, suberine and wax biosynthesis	ko00073	5/24	0.03165
	Two-component system	ko02020	5/24	0.03165
	Collecting duct acid secretion	ko04966	4/17	0.035501
	Retrograde endocannabinoid signaling	ko04723	10/71	0.041068

**Table 2 insects-13-00792-t002:** Classification of DEGs involved in host energy and nutrient metabolism.

Category	Sequence ID	Gene Name	DEGs (log2 Value)	*p* Value
Oxidative phosphorylation	BMSK0000124	probable DH dehydrogenase	−3.11178	6.97 × 10^−14^
	BMSK0000321	DH-ubiquinone oxidoreductase subunit 8	−4.38754	1.23 × 10^−23^
	BMSK0000411	DH-ubiquinone oxidoreductase B18 subunit	−11.2037	2.38 × 10^−15^
	BMSK0000424	cytochrome b-c1 complex subunit Rieske	−4.3818	2.22 × 10^−29^
	BMSK0000434	V-type proton ATPase 116 kDa subunit a isoform 1-like	−5.4141	1.12 × 10^−21^
	BMSK0000861	cytochrome c oxidase subunit 7A1	−1.37335	1.76 × 10^−5^
	BMSK0000858	cytochrome c oxidase subunit 7C	−11.654	4.86 × 10^−13^
	BMSK0000687	ATP synthase subunit gamma	−2.69535	1.22 × 10^−11^
	BMSK0000635	succinate dehydrogenase	−5.56853	1.17 × 10^−39^
	BMSK0006733	cytochrome oxidase c subunit Vib	−4.17749	6.85 × 10^−25^
	BMSK0005384	probable DH dehydrogenase	−5.3812	6.98 × 10^−30^
	BMSK0003855	cytochrome c oxidase subunit 5B	−5.06134	1.02 × 10^−33^
	BMSK0002658	DH dehydrogenase [ubiquinone] iron-sulfur protein 3	−4.08797	9.99 × 10^−19^
	BMSK0002109	DH-ubiquinone oxidoreductase 49 kDa subunit	−4.49602	1.29 × 10^−23^
	BMSK0008624	cytochrome c oxidase subunit 6A2	−4.38976	8.49 × 10^−38^
	BMSK0007592	cytochrome b-c1 complex subunit 8-like	−4.60053	3.50 × 10^−35^
Nitrogen metabolism	BMSK0000310	glutamine synthetase 2 cytoplasmic isoform X2	1.236855	4.63 × 10^−5^
	BMSK0004749	carbonic anhydrase 1 isoform X1	−4.69514	2.09 × 10^−23^
	BMSK0004746	putative carbonic anhydrase	−4.88355	8.85 × 10^−16^
Amino acid metabolism	BMSK0006194	alanine aminotransferase 1	9.384746	1.72 × 10^−71^
	BMSK0005711	inducible nitric oxide synthase-like protein	2.269884	2.81 × 10^−11^
	BMSK0005320	glutamine synthetase 1	−9.03892	4.01 × 10^−9^
	BMSK0003612	glutamate dehydrogenase	−4.376969	2.28 × 10^−25^
	BMSK0007733	malate dehydrogenase 1	−5.259740	3.16 × 10^−41^
Carbohydrate metabolism	BMSK0005441	multiple inositol polyphosphate phosphatase 1	−1.011	0.001744
	BMSK0004863	pyruvate kinase-like isoform X4	−3.59947	7.08 × 10^−10^
	BMSK0000610	aldose 1-epimerase isoform X1	−4.00298	1.71 × 10^−32^
	BMSK0000507	enolase	−4.57115	9.42 × 10^−36^
	BMSK0014846	alcohol dehydrogenase	−4.72435	2.26 × 10^−33^
	BMSK0014852	1,5-anhydro-D-fructose reductase	−3.14956	3.56 × 10^−12^
	BMSK0003272	malate dehydrogenase isoform X1	−1.35844	1.10 × 10^−5^
	BMSK0004863	pyruvate kinase-like isoform X4	−3.59947	7.08 × 10^−10^
	BMSK0004862	pyruvate kinase, alpha/beta domain	−4.3419	4.16 × 10^−12^
	BMSK0001495	glyoxylate reductase/hydroxypyruvate reductase	−2.55827	3.15 × 10^−14^
	BMSK0012704	glutamine synthetase 2 cytoplasmic-like	−6.05855	2.51 × 10^−28^
	BMSK0000686	glycine cleavage system H protein-like	−3.09021	1.93 × 10^−17^
	BMSK0000636	succinate dehydrogenase cytochrome b560 subunit	−6.39877	6.41 × 10^−21^
	BMSK0004075	chitinase isoform X1	1.200746	0.000638
	BMSK0004232	beta-N-acetylglucosaminidase 2 precursor	1.370167	4.74 × 10^−5^
	BMSK0007975	glucosamine-6-phosphate isomerase isoform X1	1.569734	4.96 × 10^−7^
	BMSK0006682	cysteine sulfinic acid decarboxylase	5.528822	1.15 × 10^−26^
	BMSK0016012	UDP-glycosyltransferase UGT33D8	1.353298	0.000243
	BMSK0014767	uridine diphosphate glucosyltransferase	6.283757	1.71 × 10^−28^
Lipid metabolism	BMSK0009526	fatty acid synthase	−2.72055	3.34 × 10^−13^
	BMSK0009516	acyl transferase domain	−2.04429	1.07 × 10^−9^
	BMSK0013704	fatty acyl-CoA reductase wat-like	−2.961012	1.66 × 10^−10^
	BMSK0004409	gamma-glutamyl transpeptidase isoform X1	−3.172825	5.65 × 10^−21^
	BMSK0004474	lysophospholipid acyltransferase 7 isoform X2	−5.22114	2.42 × 10^−23^
	BMSK0007012	non-lysosomal glucosylceramidase	−2.26115	9.35 × 10^−10^
	BMSK0013693	fatty-acyl CoA reductase 2	−1.07664	0.003835
	BMSK0005884	UDP-glycosyltransferase UGT48C1 precursor	2.373975	7.55 × 10^−7^
	BMSK0009441	phospholipase A2-like	2.619790	1.01 × 10^−14^
	BMSK0003963	uridine diphosphate glucosyltransferase precursor	6.808911	5.44 × 10^−42^

**Table 3 insects-13-00792-t003:** Summary of development-related DEGs induced by *E. japonica* parasitism.

Sequence ID	Gene Name	DEGs (log2 Value)	*p* Value
BMSK0005887	facilitated trehalose transporter Tret1-like	3.53769	2.27 × 10^−8^
BMSK0014493	juvenile hormone esterase-like isoform X2	2.9759	3.63 × 10^−8^
BMSK0014862	ecdysteroid-phosphate phosphatase	1.958563	2.08 × 10^−9^
BMSK0013050	juvenile hormone binding protein an-0128 precursor	1.777608	2.12 × 10^−6^
BMSK0010481	ecdysteroid-regulated 16 kDa protein	−1.74256	3.15 × 10^−7^
BMSK0013317	hemolymph juvenile hormone binding protein precursor	−1.01676	0.000499
BMSK0008902	juvenile hormone binding protein brP-1649 precursor	−3.73864	2.83 × 10^−10^

**Table 4 insects-13-00792-t004:** Summary of DEGs involved in cellular immune response of the parasitized silkworm.

Sequence ID	Gene Name	DEGs (log2 Value)	*p* Value
BMSK0014004	hemolin isoform X1	2.879532	1.24 × 10^−9^
BMSK0005301	hemocytin	−1.54542	1.14 × 10^−6^
BMSK0015652	scavenger receptor type C precursor	−1.55632	3.58 × 10^−6^
BMSK0013731	scavenger receptor class B member 1 isoform X2	−1.69213	1.99 × 10^−7^
BMSK0002618	very low-density lipoprotein receptor	1.666729	3.83 × 10^−8^
BMSK0001793	integrin beta4	1.163103	0.001039
BMSK0001792	integrin beta3	−1.78043	9.31 × 10^−5^
BMSK0008195	intraflagellar transport protein 46 homolog isoform X3	−3.71239	3.53 × 10^−12^
BMSK0001621	dynein intermediate chain 3	−4.69828	1.14 × 10^−22^
BMSK0005812	tetratricopeptide repeat protein 30A	−3.65764	1.69 × 10^−14^
BMSK0007120	cytoplasmic dynein 2 light intermediate chain 1 isoform X1	−3.93271	1.86 × 10^−17^
BMSK0014572	intraflagellar transport protein 20 homolog	−2.88381	2.26 × 10^−11^
BMSK0009828	dynein assembly factor 5	−2.04465	5.92 × 10^−10^
BMSK0014854	dynein beta chain	−2.94315	1.05 × 10^−20^
BMSK0011063	dynein intermediate chain 2	−4.95565	6.14 × 10^−33^
BMSK0015762	tektin-4	−4.38482	6.12 × 10^−29^
BMSK0015667	heat shock protein 83	1.229242	6.17 × 10^−7^
BMSK0009364	centromere protein J	−1.20679	6.56 × 10^−5^
BMSK0002250	intraflagellar transport protein 80 homolog isoform X1	−3.18285	1.22 × 10^−14^
BMSK0000038	actin-85C	−3.57324	1.38 × 10^−21^
BMSK0009907	cytoplasmic A3	1.491502	6.72 × 10^−9^
BMSK0015598	tubulin beta chain isoform X1	−1.04303	0.004396
BMSK0009003	beta-tubulin	−4.27001	6.91 × 10^−44^
BMSK0015598	tubulin beta chain	−4.74837	4.3 × 10^−45^
BMSK0000091	tubulin alpha-1 chain	−4.34691	9.12 × 10^−44^
BMSK0003474	tektin-B1	−4.70582	6.68 × 10^−38^

**Table 5 insects-13-00792-t005:** Humoral-immune-response-related genes.

Sequence ID	Gene Name	DEGs (log2 Value)	*p* Value
BMSK0009350	peptidoglycan-recognition protein LB-like	11.52585	1.75 × 10^−15^
BMSK0004848	peptidoglycan recognition protein S2	6.973384	2.01 × 10^−58^
BMSK0009349	peptidoglycan recognition protein S6 precursor	6.245274	1.51 × 10^−47^
BMSK0006299	beta-1,3-glucan recognition protein 3 isoform X2	1.40262	1.33 × 10^−5^
BMSK0012017	serine protease 7 precursor	7.447781	5.12 × 10^−51^
BMSK0009527	thioesterase domain	−2.45384	1.44 × 10^−10^
BMSK0012018	serine protease snake	5.39302	3.76 × 10^−39^
BMSK0015991	serpin 5	1.5216	2.88 × 10^−6^
BMSK0008651	serine protease inhibitor 6 isoform X1	5.430386	2.04 × 10^−42^
BMSK0003816	serine protease inhibitor 12 isoform X1	1.555923	1.16 × 10^−6^
BMSK0003812	serine protease inhibitor 3 isoform X1	1.338882	3.74 × 10^−5^
BMSK0003441	angiotensin-converting enzyme	−1.99032	6.07 × 10^−7^
BMSK0015864	lysozyme-like	1.688813	2.42 × 10^−8^
BMSK0013244	moricin	6.174524	1.58 × 10^−51^
BMSK0009812	gloverin 2 isoform X1	4.685764	1.64 × 10^−35^
BMSK0003513	cecropin A	3.788939	8.94 × 10^−26^
BMSK0016018	gloverin 4 precursor	3.367737	2.31 × 10^−21^
BMSK0003511	cecropin family	3.283748	6.29 × 10^−21^
BMSK0016016	gloverin1	3.227077	4.27 × 10^−20^
BMSK0016017	gloverin-like protein	2.986507	5.72 × 10^−15^
BMSK0015407	antibacterial peptide enbocin 2 precursor	2.910463	3.97 × 10^−17^
BMSK0003627	attacin-like	2.88477	2.35 × 10^−17^
BMSK0015405	antibacterial peptide enbocin 3 precursor	2.879573	7.74 × 10^−17^
BMSK0005463	ebocin 3	2.729055	6.01 × 10^−16^
BMSK0015969	gloverin 3 isoform X1	2.474034	2.23 × 10^−13^
BMSK0001742	a pirk homolog	1.564452	6.54 × 10^−7^
BMSK0006299	beta-1,3-glucan recognition protein 3 isoform X2	1.40262	1.33 × 10^−5^
BMSK0012472	eukaryotic initiation factor 4E-1	1.420455	6.20 × 10^−5^
BMSK0002354	signal transducing adapter molecule 2	1.034003	0.000234
BMSK0000175	epidermal growth factor receptor isoform X2	−1.20559	0.000214
BMSK0003517	heat shock protein 68	−4.28148	4.89 × 10^−34^
BMSK0007712	cAMP-dependent protein kinase catalytic subunit alpha-like	−5.71941	1.47 × 10^−41^
BMSK0007713	protein kinase domain	−4.55635	2.52 × 10^−27^
BMSK0015669	heat shock protein 70	4.108944	1.04 × 10^−23^
BMSK0015756	growth arrest and DNA-damage-inducible protein GADD45 alpha	1.222944	0.00032
BMSK0001919	protein 60A	1.695163	3.1 × 10^−7^
BMSK0001708	Tgif2	−5.366	9.01 × 10^−45^

## Data Availability

The RNA-Seq data were deposited in NCBI with BioProject ID: PRJNA871044.

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
