# Peer review of "Parasitism by the Tachinid Parasitoid Exorista japonica Leads to Suppression of Basal Metabolism and Activation of Immune Response in the Host Bombyx mori"

_insects, 2022, doi:10.3390/insects13090792_

Round 1

Reviewer 1 Report

This Manuscript is an interesting study of physiological-genetic analysis in which its results demonstrate that the tachinid parasitoid Exorista japonica perturbs the basal metabolism and induces the costly immunity of the host, in this case study Bombyx mori, thus leading to incomplete larval-pupal ecdysis of the host.  In brief, the manuscript provided insights into how tachinid parasitoids can modify host basal metabolism and immune response for the benefit of developing parasitoid larvae. I found this a well-designed and performed study; goals were concurrent with the results presented; data were clearly presented and properly analyzed. The manuscript is clear and well written. From my point of view. The English language are fine, but the drafting style would need revision, although I am not a native English speaker; I suggest a more thorough review of the English by someone proficient in the language.  I think this study provides valuable information on the effect of parasitoids on the immune system of their hosts. I do not see any flaws with the current version. I suggest minor changes, more of a drafting style, for example, In Abstract (line 22), the author wrote: “The dipteran tachinid parasitoids that are important biocontrol agents….”; I suggest writing it as follows: “Tachinid parasitoids are important biocontrol agents…”. In general, throughout the manuscript there are very long sentences that can be simplified by using fewer words. However, this comment does not detract from the excellent study carried out by the authors, which deserves to be published in the prestigious INSECTS journal.

Reviewer 2 Report

The present manuscript provides a very interesting story. It is well conducted, with clear results and documented discussion. It is well written in general, with some issues which have to be clarified/added/changed prior to the final decision. Detailed info can be found in the attached PDF file.

Reviewer 3 Report

The article "The tachinid parasitoid Exorista japonica parasitism leads to suppression of basal metabolism and activation of immune response in the host Bombyx mori" summarize the effects of parasitoid Exorista japonica to Bombyx mori from transcriptomic point of view. The aim to use transcriptomic analysis to discover influence of parasitoid to host is interesting and potentially can lead to many significant discoveries. However, the authors stopped with their explanations on basic levels as disrupted metabolism and increased immune reactions. It is true that authors provide first transcriptomic analysis of mentioned species; nevertheless, there is several studies concerning parasitic relationship showing the same pattern of disrupted metabolic processes and immune activation. It would be of more interests to show some specific response of host to the parasitic fly to distinguish this study from other already published ones. The presented result can be used as cornerstone for interesting studies like what are the specific mechanisms driving the cellular immune reactions since the Hemocytin steadily increase activity during the infection? Or what leads to such a high upregulation of Cactus in 7 DAP when all the other factors are so low? The authors state this necessity of mechanistic studies in the discussion and conclusion, but it would highly elevate current manuscript if at least one of the mechanism would be included.

Introduction:

The introduction covers all the necessary information relevant for the study. It is perfectly written with easy legibility and strong flow.

Material and methods

In the section 2.2 is only description of how the samples were collected on 4 DAP and that this samples were used for transcriptomics analysis. However, there is no description of how the samples were collected for the other time points used in the study. There are 3-8 DAP samples used for PCR analysis but not the description how they were obtained.

In section 2.3 the authors describe the transcriptomic analysis however they do not provide the original dataset uploaded to publicly available repository. This is part of good publishing practice and the access to the data should be allowed.

In the section 2.6 the authors describe their usage of Student’s t test as main statistical analysis with assumption that all the results were normally distributed. However, in the study is used comparison of multiple genes to control per one time point. For such design the ANOVA one-way analysis with subsequent multiple comparison post-hoc test like Bonferroni is more suitable and recommended way. 

Results

In part 3.1 line 195-198 the authors state that they used ten genes to confirm accuracy of the transcriptomic analysis. However, they do not tell why they selected those ten genes. Is there some reasoning why those genes were selected?

In the figure 1 description the letters do not correspond with those stated in the figure itself. Also, it would be beneficial to explain what does the “P” and “NP” stands for in the figure 1B same as to provide some more details about that analysis since it is poorly described even in the correspond result section.

In section 3.2 I would appreciate more details included about the analysis like what are the p-values and number of genes in the respective GO terms.

Figure 2 it would be beneficial to highlight which GO terms belong to “biological process”, “cellular component” and “molecular function” respectively. Also, what does the top x-axis with percentage means?

For the whole part with PCR analysis authors compare the gene expression only to house keeping gene from the same set of parasitised samples. However, more relevant biological option would be comparison to unparasitised sample of the same developmental stage. In that case it would be possible to spot if and how much the presence of parasitoid influence the reactions of host. This is one of the crucial points which should be resolved for successful publication.

Line 338 there are both forms of to be verb.

Line377-380 is stated that all antimicrobial peptides are upregulated at all time-points. However, according to figure 4B Moricin is not significantly up-regulated at 3DAP.

Discussion

Line 424-426 is there some study showing that parasitoid feeds primarily on the fat body?

Line 464-471 is it possible to check that the encapsulation is disrupted in the host? Or at least provide some resource which checked the stated hypothesis?

Line 487 it should be “immunity” instead of “immune”

Line 499 and 500 should be “JAK/STAT” instead of “JAK/ATAT” 

Author Response

The article "The tachinid parasitoid Exorista japonica parasitism leads to suppression of basal metabolism and activation of immune response in the host Bombyx mori" summarize the effects of parasitoid Exorista japonica to Bombyx mori from transcriptomic point of view. The aim to use transcriptomic analysis to discover influence of parasitoid to host is interesting and potentially can lead to many significant discoveries. However, the authors stopped with their explanations on basic levels as disrupted metabolism and increased immune reactions. It is true that authors provide first transcriptomic analysis of mentioned species; nevertheless, there is several studies concerning parasitic relationship showing the same pattern of disrupted metabolic processes and immune activation. (1) It would be of more interests to show some specific response of host to the parasitic fly to distinguish this study from other already published ones. (2) The presented result can be used as cornerstone for interesting studies like what are the specific mechanisms driving the cellular immune reactions since the Hemocytin steadily increase activity during the infection? (3) Or what leads to such a high upregulation of Cactus in 7 DAP when all the other factors are so low? (4) The authors state this necessity of mechanistic studies in the discussion and conclusion, but it would highly elevate current manuscript if at least one of the mechanism would be included.
Response:
(1) We thank you for raising this issue and we have discussed the specific and conserved responses of the host when parasitized by fly parasitoids or other parasitoid species such as wasps in lines 454-463.
(2) Thank you for the valuable suggestions. Hemocytin is expressed in granules of silkworm blood cells and can recognize pathogens and initiate host immune response. It can initiate host encapsulation and melanization pathway to seal wounds and envelop and destroy pathogens. In this study, hemocytin was down-regulated at 3-, 4-, 5-DAP, and up-regulated at 7-, 8-DAP (Figure 4A), we also found that host melanization was activated at 4-DAP, it is possible that the fly parasitoid inhibits hemocytin expression via a mechanism to regulate the level of hemolymph melanization to protect itself from being killed by the host.
(3) Cactus is negative regulator in Toll pathway, its expression was not altered at relative early infection stage, but up-regulated at late infection stage (7- and 8-DAP), concomitantly, the positive regulator Spatzle was down-regulated at early (3-DAP) and late infection stage (8-DAP), suggesting that Toll pathway was inhibited during parasitoid infection.
(4) We have found a lot of interesting results in this manuscript that can be further mechanistically studied, the results in this study provide basis for future studies.

Introduction:
The introduction covers all the necessary information relevant for the study. It is perfectly written with easy legibility and strong flow.
Response: Thank you for the positive comment from this reviewer.

Material and methods
In the section 2.2 is only description of how the samples were collected on 4 DAP and that this samples were used for transcriptomics analysis. However, there is no description of how the
samples were collected for the other time points used in the study. There are 3-8 DAP samples used for PCR analysis but not the description how they were obtained.
Response: Thank you for pointing out this, we have added the description of sample collection for all time points that used for transcriptomics and RT-qPCR analysis in lines 128 and 129.

In section 2.3 the authors describe the transcriptomic analysis however they do not provide the original dataset uploaded to publicly available repository. This is part of good publishing practice and the access to the data should be allowed.
Response: Thank you for your advice, we have deposited the RNA-Seq data into the public SRA database with BioProject ID: PRJNA871044 and we added this information in the revised text.

In the section 2.6 the authors describe their usage of Student’s t test as main statistical analysis with assumption that all the results were normally distributed. However, in the study is used comparison of multiple genes to control per one time point. For such design the ANOVA one-way analysis with subsequent multiple comparison post-hoc test like Bonferroni is more suitable and recommended way.
Response: We have adopted the method of ANOVA one-way analysis followed by Tukey’s test at P < 0.05 to do statistical analysis and revised all results in the revised version.

Results
In part 3.1 line 195-198 the authors state that they used ten genes to confirm accuracy of the transcriptomic analysis. However, they do not tell why they selected those ten genes. Is there some reasoning why those genes were selected?
Response: These ten genes are mainly involved in immune response and nutrient metabolism, we have added the explanation in lines 199 and 200.

In the figure 1 description the letters do not correspond with those stated in the figure itself. Also, it would be beneficial to explain what does the “P” and “NP” stands for in the figure 1B same as to provide some more details about that analysis since it is poorly described even in the correspond result section.
Response: We have corrected it in the figure legend for Figure 1 and added more details of Figure 1B in lines 186 and 188.

In section 3.2 I would appreciate more details included about the analysis like what are the p-values and number of genes in the respective GO terms.
Response: Thanks for the advice. In our study, we used GO analysis to classify the transcripts into the three functional groups, and the results were shown in Figure 2.
In fact, we also performed GO enrichment analysis, however, most of the DEGs were enriched into cell projection and motor activity. Since we did not get much information from GO enrichment analysis and thus did not present the results in the text. Meanwhile, we have obtained more interesting information via KEGG pathway enrichment analysis in section 3.2. We also added the input/background number in the respective KEGG terms in Table 1.

Figure 2 it would be beneficial to highlight which GO terms belong to “biological process”, “cellular component” and “molecular function” respectively. Also, what does the top x-axis with percentage means?
Response: We have added the GO terms to Figure 2. The upper top x-axis represents the proportion of the percentage of the number of genes corresponding to the function, and the lower x-axis represents the number of genes corresponding to the function. And we have added this information in lines 245-248.

For the whole part with PCR analysis authors compare the gene expression only to housekeeping gene from the same set of parasitised samples. However, more relevant biological option would be comparison to unparasitised sample of the same developmental stage. In that case it would be possible to spot if and how much the presence of parasitoid influence the reactions of host. This is one of the crucial points which should be resolved for successful publication.
Response: Thank you for the suggestion. In the revised text, to make it easier to recognize the parasitoid influence on the reactions of host, we presented the expression levels of genes relative to housekeeping genes in control and parasitized groups in Figures 3 and 4.

Line 338 there are both forms of to be verb.
Response: We have revised it.

Line377-380 is stated that all antimicrobial peptides are upregulated at all time-points. However, according to figure 4B Moricin is not significantly up-regulated at 3DAP.
Response: Thank you for pointing out this, we have corrected it.

Discussion
Line 424-426 is there some study showing that parasitoid feeds primarily on the fat body?
Response: Thank you for raising this concern. Valigurová et al have demonstrated that the tachinid larvae burrow through the host integument after hatching and start feeding mostly on host hemolymph and the fat body. But they didn’t show the direct evidence of this phenomenon, so we deleted this sentence for accurate description in revised version.
Reference: Valigurová, A.; Michalková, V.; Koník, P.; Dindo, M. L.; Gelnar, M.; Vaňhara, J., Penetration and encapsulation of the larval endoparasitoid exorista larvarum (diptera: tachinidae) in the factitious host galleria mellonella (lepidoptera: pyralidae). B. Entomol. Res. 2014, 104(02), 203-212. DOI:10.1017/S0007485313000655

Line 464-471 is it possible to check that the encapsulation is disrupted in the host? Or at least provide some resource which checked the stated hypothesis?
Response: Yes, it is possible. Many researchers have hypothesized that encapsulation in host is induced, and the tachinid larvae can survive from host immune response by moulding the host encapsulation into respiratory funnels (as shown in Figure 1A) to avoid be suffocated. However, until now, there is no direct evidence that encapsulation occurs during parasitoid fly larva infection. In our study, we only deduced that host cellular response was modulated by parasitoid infection, we did not know how host encapsulation occurred and was disrupted.

Line 487 it should be “immunity” instead of “immune”
Response: We have replaced “immune” with “immunity”.

Line 499 and 500 should be “JAK/STAT” instead of “JAK/ATAT”
Response: We have revised it.